

# Higher central fat and poor self-body image in short-stature overweight/obese women living in Brazilian shantytowns

Nassib Bezerra Bueno[1], Telma Toledo Florêncio[1], Fabiana Albuquerque Cavalcante[1], Isabela Lopes Lins[1], Ana Grotti Clemente[1] and Ana Lydia Sawaya[2]

[1] Faculdade de Nutrição, Universidade Federal de Alagoas, Maceió, AL, Brazil
[2] Departamento de Fisiologia, Universidade Federal de São Paulo, São Paulo, SP, Brazil

## ABSTRACT

**Background:** Short stature in adult life, a possible consequence of poor perinatal conditions, is associated with higher risk of mortality and social disabilities. We aimed to determine whether low-income, overweight/obese, short-stature (SS) women show alterations in body composition, self-body-image perception, and biochemical profile compared to their non-short (NS) counterparts.

**Methods:** A cross-sectional study was conducted with women living in shantytowns and mother or relatives to undernourished children treated in a center for recuperation and nutritional education. Inclusion criteria were: (1) age, 19–45 years; (2) stature < 152.3 cm or > 158.7 cm; and (3) body mass index > 25 kg/m$^2$. Socioeconomic, anthropometric, biochemical, and body image data were collected. We analyzed 56 SS and 57 NS women.

**Results:** The SS group showed a higher waist-to-height ratio (WHtR) (mean: 0.63; standard deviation: 0.06 for SS and mean: 0.60; standard deviation: 0.07 for the NS group; p = 0.02), and, in the adjusted analysis, showed lower fat-free mass (Estimated Marginal Mean for the SS group: 45.7 kg 95% confidence intervals (CI) (45.2–46.2) and for the NS group: 46.9 kg 95% CI (46.4–47.4); p < 0.01) and higher fat mass (Estimated Marginal Mean for the SS group: 32.5 95% CI (31.9–33.0) and for the NS group: 31.4 kg 95% CI (30.9–31.9); p < 0.01). Body mass index was a better predictor of current self-body-image perception for NS women. The SS coefficient values were $\beta = 0.141$, SE = 0.059, and R$^2$-Nagelkerke = 0.107, and the NS coefficients values were $\beta = 0.307$, SE = 0.058, and R$^2$-Nagelkerke = 0.491 (Z = 2.006; p < 0.05). Considering the obese subgroup, six out of 32 (18.8%) SS women and 14 out of 33 (42.4%) NS women perceived themselves as obese ($\chi^2 = 4.27$; p = 0.03). This difference remained significant even after adjustment by age, schooling, and number of children (p = 0.04). Only the total thyroxin showed significant differences between groups, lower in SS women (p = 0.04).

**Discussion:** Overweight/obese, low-income SS women have more central adiposity and impaired self-body image perception, and the body mass index is a weaker predictor of it, compared to NS women. Misperception about body size may be linked with an overestimation of health and underestimation of risk, which may lead to a lower utilization of the health care system and inadequate physician counseling. These features may account, at least partially, for the higher mortality risk seen in SS adults.

Corresponding author
Nassib Bezerra Bueno,
nassibbb@hotmail.com

# INTRODUCTION

Studies on the developmental origins of health and disease aim to understand how events in early life shape the morbidity risk later in life (*Gluckman, Hanson & Buklijas, 2010*). As non-communicable chronic diseases are currently one of the major causes of morbidity and mortality worldwide and their prevalence is currently increasing, the number of studies being conducted on this topic has also been increasing. Poor health in the early stages of life may be responsible for long-term social and metabolic/morphologic consequences (*Victora et al., 2008*) and is responsible for the intergenerational transmission of inequality, where disadvantaged parents give birth to disadvantaged children who will probably be disadvantaged parents in the future (*Aizer & Currie, 2014*).

Adult short stature is a direct consequence of impaired early growth caused by poor health, particularly inadequate nutrition and recurrent infections (*Shrimpton et al., 2001*). It is possible to attenuate adult short stature with compensatory growth during childhood. However, in low- and middle-income countries, people tend to remain in the setting in which they developed childhood undernutrition, leading to poor compensation of growth failure and, ultimately, the children grow into short adults (*Martorell, Khan & Schroeder, 1994*). An analysis of a Brazilian birth cohort showed that women who had a height-for-age $Z$ score at 2 years lower than $-3$, presented an attained height of approximately 146 cm in adulthood, in contrast with 163 cm presented by those who had a height-for-age $Z$ score greater than $-1$ (*Victora et al., 2008*). Additionally, in the analysis of five birth cohorts in developing countries, stunted children were more likely to have a reduced lean body mass, attain a lower education level, and have reduced earnings in adulthood (*Victora et al., 2008*).

Short stature in adult life is associated with a higher mortality risk. A recent individual-patient meta-analysis of more than 16.1 million person/year showed that the risk of all-cause mortality is 3% lower per 6.5 cm height in adult life, although disaggregation by cause-specific mortality revealed stronger and directionally opposing relationships with the risk of death from different major causes of chronic disease. The study concluded that taller people have a lower risk of death from coronary disease, stroke subtypes, heart failure, oral and gastric cancers, chronic obstructive pulmonary disease, mental disorders, liver diseases, and external causes (*The Emerging Risk Factors Collaboration, 2012*).

An overweight/obese short-statured (SS) adult will be at a great risk of chronic disease, especially cardiovascular diseases. Women in a low-income region from a developing country that is facing a nutrition transition are especially at risk owing to persistent weight gain due to low energy expenditure, working at home, and hyperenergetic diets (*Kanter & Caballero, 2012*). This is a common situation in Brazilian northeast.

Mechanisms that justify the greater risk experienced by SS adults are not fully understood, especially in the context of a low-income setting. Hence, the present study aimed to determine whether overweight/obese SS women from Maceió, capital city of the state of Alagoas, and one of the poorest capital city of Brazil, show any alterations in

their body composition, self-body-image perception, and blood biochemical profile as compared to their non-short (NS) counterparts.

## METHODS

### Ethical aspects

Data were collected from all participants after they provided written informed consent. The study was approved by the Ethical Research Committee of the Federal University of São Paulo, number 275184.

### Study design and subjects

This was a cross-sectional study of women living in a shantytown in Maceió-AL, Brazil. The subjects were mothers and/or female relatives of undernourished children who attended the Centre for Recuperation and Nutritional Education (CREN/Maceió), an extension of the Federal University of Alagoas that treats undernourished children with a goal of improving their nutritional status.

We investigated two different groups: SS women and NS women. Inclusion criteria were as follows: (i) age, 19–45 years; (ii) stature $\leq$ 152.3 cm (5th percentile of the World Health Organization growth curves) (SS group) or > 158.7 cm (25th percentile) (NS group) (*World Health Organization, 2007*); and (iii) a body mass index (BMI) > 25 kg/m$^2$. Pregnant and lactating women and those taking chronic medication, insulin, and antiretrovirals were not included.

We used a non-probability convenience sampling approach, inviting all mothers and relatives of the undernourished children treated at CREN to a screening at the center. All included women were then invited to join a behavioral weight-loss program.

### Data collection

Data were collected at CREN with a previously tested and structured protocol regarding personal, socioeconomic, and demographic information such as age, years of schooling, occupation, household conditions, and number of family members, wage, government benefits, precedence (city or rural).

Anthropometric measurements (body weight, stature, and waist and hip circumferences) were obtained from all participants while they were wearing light clothing without shoes. Body weight was determined using electrical bioimpedance with Tanita Body Fat analyzer (model TBF-300; Tanita Corporation of America, Inc., Arlington Heights, IL, USA), which also measured fat mass, fat-free mass, and body fat percentage and calculated the basal metabolic rate (BMR). To measure the stature, the women were asked to stand in the full upright position using a portable stadiometer, which was equipped with an inextensible metric tape marked by 0.1-cm graduations. Waist circumference (WC) was measured during a normal expiration, using an inextensible flexible metric tape at the umbilical point. Hip circumference was measured at the point yielding the maximum circumference over the buttocks. The waist-to-hip ratio (WHR) and waist-to-height ratio (WHtR) were also calculated by dividing the WC by the hip circumference, and the WC for the height in

centimeters, respectively. Body surface area was calculated according to method used by *Mosteller (1987)*. Blood pressure was measured on three consecutive occasions using calibrated monitors against mercury manometers (Model HEM-421-CO; Omron, Amsterdam, Netherlands) by the standard technique, with individuals being seated and relaxed.

To measure body image, a scale consisting of nine female figures, ranging from very thin to very obese (Figs. 1–9), was used (*Stunkard, 2000*). Participants were asked to choose one figure that, in their opinion, currently represented their body and one that represented their ideal body. The discrepancy score was calculated by subtracting the ideal self-body image number from the current self-body image number. These figures were classified into underweight (Figs. 1 and 2), normal weight (Figs. 3 and 4), overweight (Figs. 5 through 7), and obese (Figs. 8 and 9), according to the method of *Lynch et al. (2009)*. This scale has already been translated into Portuguese and tested with Brazilian women (*Scagliusi et al., 2006*).

Blood samples were collected at CREN facilities for biochemical profile studies. Women were instructed to fast overnight before undergoing venous puncture. Tests for the following parameters were performed: blood glucose and insulin, Homeostasis Model Assessment for insulin resistance (HOMA-IR), $\beta$-cell function (HOMA-$\beta$), total cholesterol, high-density lipoprotein, low-density lipoprotein, triacylglycerols, aspartate amino-transferase, alanine amino-transferase, gamma-glutamyl transpeptidase, thyroid-stimulating hormone, triiodothyronine, thyroxin, albuminuria, and cystatin-C.

## Statistical analysis

Continuous and discrete data are presented as mean and standard deviation and categorical data, as absolute and relative frequency. For all continuous variables, the normality and variance's homogeneity assumptions were tested by the Lilliefors' and Levene's tests. When the assumptions were met, Student $t$-tests were conducted; otherwise, Welch's t-tests were conducted. Discrete variables were subjected to Mann-Whitney tests. Categorical variables were compared using the $\chi^2$ test or Fisher Exact test, when appropriate. Multivariate analysis with dichotomous outcomes were performed by means of logistic regression.

Variables that could be influenced by body size (such as fat mass and lean mass) were subjected to an analysis of covariance (ANCOVA) using body weight and age as covariates, and the estimated marginal means (EMM) and 95% confidence intervals (CI) were calculated. An ordinal regression was performed using current self-body-image perception as the dependent variable and BMI as the independent variable for each subgroup (NS and SS). Further, regression coefficients were compared by Fisher's $Z$-test. In addition, a full ordinal regression model including confounders and an interaction term between height and BMI was designed. For all analyses, a two-tailed approach was adopted, and alpha was set at 5%. For the statistical analyses, SPSS v20.0 (IBM statistics, Chicago, IL, USA) was used.

## RESULTS

In total, 262 women were invited to the screening, comprising all mothers and relatives of the undernourished children treated at CREN. From these, 149 were not included in our analysis: 116 did not meet inclusion criteria for age, height and/or body mass index; 15 were pregnant or lactating; five were taking insulin; and 13 refused to participate. A total of 113 women were finally included in the analyses. Of these, 57 were NS and 56 were SS women. Socio-economic characteristics of the population are shown in Table 1. In general, the SS group received less schooling, had more children, and were more likely to arise from rural areas. The others variables did not show any significant differences.

Table 2 presents the women's anthropometric and self-body image assessments. The NS group had significant higher values for body size variables, except WC, which did not show any significance, and WHtR, which was higher in the SS group, indicating a central-obesity profile.

Table 3 shows the adjusted analysis for the between-group comparison for fat mass, lean mass, body fat percentage and BMR, including body weight as a covariate in the model. In this case, the variables for body composition showed opposite results as those of the unadjusted analysis, shown in Table 2, that is, according to the EMM of the adjusted analysis, SS women showed lower lean mass, and higher fat mass. These differences between groups were significant (Table 3). Results of the biochemical and blood pressure analyses are presented in Table 4. Only a small difference in the total thyroxin level was noted, and the SS group showed lower values than the NS group.

Although the self-body-image assessment did not show any significant differences between groups, ordinal regression analysis showed that BMI is a significant better predictor of current self-body-image perception for NS women than for SS women (Fig. 1). The SS coefficient values were $\beta = 0.141$, $SE = 0.059$, and $R^2$-Nagelkerke = 0.107, and the NS coefficients values were $\beta = 0.307$, $SE = 0.058$, and $R^2$-Nagelkerke = 0.491 ($Z = 2.006$; $p < 0.05$). In the full model, including age, schooling, and number of children, the interaction between BMI and height category was also significant ($p = 0.018$), showing that height category acts as an effect moderator variable.

When considering only the obese subgroup, six out of 32 (18.8%) SS women and 14 out of 33 (42.4%) NS women perceived themselves as obese based on the categorized score of the scale ($\chi^2 = 4.27$; $p = 0.03$). This difference remained significant even after adjustment by age, schooling, and number of children ($p = 0.04$).

Using WHtR as main variable, with an alpha-value of 5%, a mean standard deviation of 0.065 and the sample size and means reported in the study, the post-hoc power for the comparisons between groups was 68%.

## DISCUSSION

As previously outlined, the mechanisms that lead to increased cardiovascular risks in SS adults are not fully understood. Height is a biomarker that congregates the genetic endowment and early-life experiences of the individuals (NCD Risk Factor Collaboration, 2016). Therefore, the increased risk presented by short adults could be due to morphologic aspects, such as smaller coronary vessels diameter and/or faster heart

**Table 1  Socioeconomic characteristics of the sample (n = 113).**

| Variables | Short-statured women (n = 56) | | Non–short-statured women (n = 57) | | |
| | Mean or "frequency" | SD or "%" | Mean or "frequency" | SD or "%" | P-value |
| --- | --- | --- | --- | --- | --- |
| Age (years)[1] | 32.39 | 6.58 | 30.47 | 6.61 | 0.12 |
| Schooling (years)[1] | 5.64 | 3.37 | 7.61 | 3.34 | < 0.01 |
| Total wage (US$)[1] | 381.81 | 233.06 | 437.77 | 255.66 | 0.23 |
| Wage per capita (US$)[1] | 94.77 | 73.64 | 112.94 | 72.26 | 0.19 |
| Number of children[1] | 2.76 | 1.39 | 2.15 | 1.28 | 0.01 |
| Rural origin[2] | 31 | 55.3 | 12 | 21.1 | < 0.01 |
| Married/stable union[2] | 45 | 81.9 | 45 | 83.7 | 0.69 |
| Unemployed[2] | 22 | 40.0 | 19 | 35.2 | 0.51 |
| Receive government benefits[2] | 46 | 83.6 | 39 | 72.2 | 0.09 |
| Alcohol user[2] | 18 | 32.7 | 15 | 27.8 | 0.49 |
| Tobacco user[2] | 3 | 5.5 | 2 | 3.7 | 0.63 |
| House with uncoated floor[2] | 28 | 50.0 | 26 | 42.6 | 0.64 |
| House with uncoated walls[2] | 20 | 34.5 | 25 | 40.7 | 0.37 |
| Absence of sewage system[2] | 51 | 94.7 | 51 | 94.4 | 0.77 |

Notes:
[1] Results expressed as mean and standard deviation.
[2] Results expressed as absolute and relative frequency.

**Table 2  Anthropometric variables and self-body image perception of the sample (n = 113).**

| Variables | Short-statured women (n = 56) | | Non–short-statured women (n = 57) | | |
| | Mean | Standard deviation | Mean | Standard deviation | P-value[1] |
| --- | --- | --- | --- | --- | --- |
| Weight (kg) | 71.46 | 10.07 | 85.59 | 15.32 | < 0.01 |
| Height (cm) | 149.5 | 2.9 | 162.6 | 3.6 | < 0.01 |
| Body mass index (kg/m$^2$) | 31.92 | 4.38 | 32.36 | 5.74 | 0.64 |
| Body surface area (m$^2$) | 1.71 | 0.12 | 1.95 | 0.17 | < 0.01 |
| Body fat (%) | 38.00 | 5.1 | 41.62 | 4.94 | < 0.01 |
| Fat mass (kg) | 27.58 | 7.42 | 36.30 | 10.67 | < 0.01 |
| Lean mass (kg) | 43.66 | 3.41 | 49.02 | 4.98 | < 0.01 |
| Basal metabolic rate (kcal/day) | 1,463.6 | 106.5 | 1,635.0 | 155.4 | < 0.01 |
| Waist circumference (cm) | 94.18 | 10.22 | 97.90 | 12.66 | 0.09 |
| Hip circumference (cm) | 107.80 | 8.67 | 114.50 | 10.94 | < 0.01 |
| Waist/hip ratio | 0.87 | 0.08 | 0.85 | 0.06 | 0.13 |
| Waist/height ratio | 0.63 | 0.06 | 0.60 | 0.07 | 0.02 |
| Current self-body image | 6.25 | 1.28 | 6.50 | 1.50 | 0.32 |
| Ideal self-body image | 3.45 | 1.22 | 3.75 | 0.99 | 0.14 |
| Discrepancy score | 2.80 | 1.55 | 2.75 | 1.59 | 0.86 |

Note:
[1] P-value for the "t" tests for independent samples.

**Table 3** Analysis of the anthropometric variables and basal metabolic rate adjusted by body weight (n = 113).

| Variables | Short-statured women (n = 56) | | Non–short-statured women (n = 57) | | |
| --- | --- | --- | --- | --- | --- |
| | Estimated marginal mean | 95% confidence interval | Estimated marginal mean | 95% confidence interval | P-value[1] |
| Fat mass (kg) | 32.5 | 31.9–33.0 | 31.4 | 30.9–31.9 | < 0.01 |
| Lean mass (kg) | 45.7 | 45.2–46.2 | 46.9 | 46.4–47.4 | < 0.01 |
| Body fat (%) | 40.4 | 5.0 | 39.4 | 4.9 | 0.06 |
| Basal metabolic rate (kcal) | 1,534.9 | 1,525–1,543 | 1,565 | 1,556–1,573 | < 0.01 |

**Note:**
[1] P-value for the ANCOVA using group as fixed fator and body weight as a covariate.

**Table 4** Blood biochemical markers and blood pressure values of the sample (n = 113).

| Variables | Short-statured women (n = 56) | | Non–short-statured women (n = 57) | | |
| --- | --- | --- | --- | --- | --- |
| | Mean | Standard deviation | Mean | Standard deviation | P-value[2] |
| Glucose (mmol/L)[1] | 4.21 | 0.44 | 4.39 | 0.75 | 0.13 |
| Total cholesterol (mmol/L) | 5.0 | 1.03 | 4.79 | 0.88 | 0.25 |
| HDL-cholesterol (mmol/L)[1] | 1.24 | 0.20 | 1.26 | 0.27 | 0.66 |
| LDL-cholesterol (mmol/L) | 3.09 | 0.84 | 2.83 | 0.75 | 0.08 |
| Triacylglycerols (mmol/L) | 1.45 | 0.78 | 1.53 | 0.77 | 0.62 |
| Alanine amino-transferase ($\mu$kat/L)[1] | 0.33 | 0.15 | 0.41 | 0.31 | 0.10 |
| Aspartate amino-transferase ($\mu$kat/L) | 0.41 | 0.10 | 0.43 | 0.21 | 0.53 |
| Gamma-glutamyl transferase ($\mu$kat/L)[1] | 0.59 | 0.26 | 0.69 | 0.44 | 0.15 |
| Albuminuria (mg/g cr) | 9.06 | 6.09 | 9.96 | 10.34 | 0.57 |
| Cystatin-C (mg/L) | 0.67 | 0.11 | 0.68 | 0.11 | 0.70 |
| Thyroid-stimulating hormone (mIU/L) | 1.66 | 0.87 | 1.74 | 0.34 | 0.67 |
| Total-T3 (nmol/L) | 1.83 | 0.37 | 1.91 | 0.43 | 0.27 |
| Total-T4 (nmol/L) | 120.2 | 22.77 | 128.4 | 21.49 | 0.04 |
| Insulin (pmol/L)[1] | 57.23 | 11.32 | 67.16 | 12.15 | 0.09 |
| HOMA-%B | 156.26 | 64.16 | 163.81 | 70.79 | 0.55 |
| HOMA-IR[1] | 1.15 | 0.53 | 1.41 | 0.98 | 0.08 |
| Systolic blood pressure (mmHg) | 123.27 | 15.21 | 122.65 | 12.92 | 0.81 |
| Diastolic blood pressure (mmHg) | 76.75 | 11.30 | 74.70 | 10.06 | 0.31 |

**Notes:**
HDL, high-density lipoprotein; LDL, low-density lipoprotein; HOMA-%B, Homeostasis Model Assessment for $\beta$-cell function; HOMA-IR, Homeostasis Model Assessment for insulin resistance; T3, triiodothyronine; T4, thyroxine.
[1] Variables were subjected to Welch's t-test, due to unequal variances.
[2] P-value for the independent samples Student t-test, unless otherwise stated.

rate (*Smulyan et al., 1998*). Nevertheless, as SS in adult life is believed to reflect, at least in part, poor nutrition and socioeconomic aspects in early-life, the possibility of a "metabolic programming" is plausible and accepted nowadays as the theory of "developmental origins of health and disease" (*Heindel et al., 2015*). This theory states that adverse events in early-life (that would impair growth and possibly lead to SS in adult life)

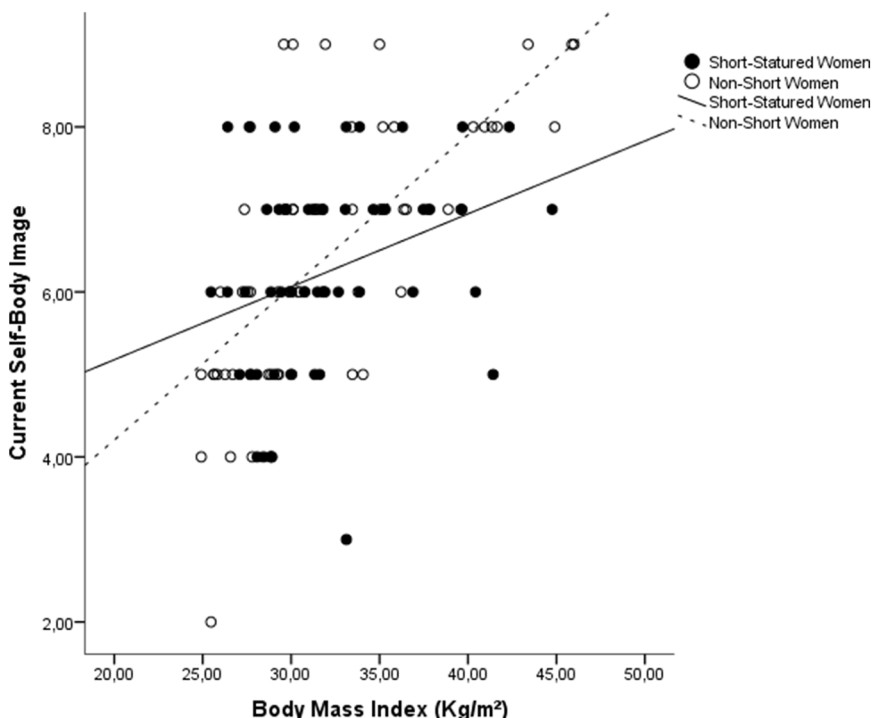

**Figure 1 Scatterplot of the influence of BMI on current self-body image, stratified by groups.** Slopes differ signifcantly between groups ($Z = 2.006$; $p < 0.05$).

could contribute to increased risk of cardiovascular disease in adulthood. In our cross-sectional study, we found that overweight/obese SS women had relatively more fat mass and less lean mass than their NS counterparts. In addition, the BMI of SS women had significantly less power to predict their actual body image. We believe that these features may account, at least partially, for the higher mortality risk seen in SS adults.

Our two study groups were socially very similar to each other, except for their schooling, number of children and precedence. In populations originating from rural areas, where the health conditions are worse, delivery at home is usually expected, and it is likely that the mothers came to the capital city looking for better life conditions, which is a common phenomenon in this population. As a whole, Alagoas is a very poor Brazilian state. A probabilistic survey conducted in 1992 showed that stunting levels among children were 22.1% (*Ferreira et al., 2013*). These levels were even higher in a survey with the rural population in 1995 reaching 39.8% (*Ferreira et al., 1997*). Hence, it is likely that our sample, specially the SS group, went through hazardous conditions during early life.

Undernutrition early in life has been consistently linked with a lower educational level (*Grantham-McGregor et al., 2007*). Lower education, in turn, is linked with household crowding (*Melki et al., 2004*), which we indirectly measured by the number of children in the household; this is a known form of chronic stress that is detrimental to the psychological well-being (*Fuller et al., 1996*). The increased adiposity noted in our study confirmed the trend reported by *López-Alvarenga et al. (2003)*, although they had analyzed both men and women, and not all were overweight/obese.

Our finding is also in accordance with that of *Martins et al. (2004)* who reported that stunted girls gained less lean mass and had a significantly higher fat mass at follow-up than their baseline values. We analyzed body composition data by using an ANCOVA to consider the differences in body size between groups, since NS women are significantly bigger than SS women. We chose to use body weight as a covariate in the model because stature was already considered as a fixed factor in the analysis (the subjects were grouped based on stature); hence, using body surface area would use stature twice in the model.

Increased adiposity, which is associated with central obesity (*Zhang et al., 2007*), is a well-known mortality risk for women and is of a greater risk to young women (*Hu et al., 2004*; *Lahmann et al., 2002*). In our sample, we did not find any differences in the WHR between the NS and SS groups, as was also reported by *Florêncio et al. (2007)*. Nevertheless, we did find differences in the WHtR, which is known to be as good as the other markers of central adiposity that predicted cardiovascular events in women (*Page et al., 2009*) and a better marker than WHR that predicted albuminuria in diabetic Chinese women (*Tseng, 2005*). Indeed, the utility of WHR, although used widely, is questionable because individually, waist and hip circumferences are known to have contrasting associations with mortality (*Bigaard et al., 2004*). In our sample, although NS women had a significantly greater WC than SS women, when the WC values were divided by stature, the NS women had a greater anthropometric advantage. Still regarding the WC, as we measured it at the umbilical point, and not in the midpoint between iliac crest and lower rib, it is possible that the value of WC for some women, especially those with higher adiposity were overestimated.

*Florêncio et al. (2007)* reported that overweight/obese SS women had a greater prevalence of insulin resistance and altered lipid profile as compared to women with average stature. However, we were unable to confirm these findings. Nevertheless, they assessed women in a poorer socioeconomic status, older and with lower stature than ours. Interestingly, although the level of thyroxin was lower in the SS group, no effects were seen in the levels of thyroid-stimulating hormone even after log-transformation and removal of outliers. This lower thyroxin level could explain the lower BMR found in the adjusted analysis. However, when considering the absence of differences in thyroid-stimulating hormone between groups, and the fact that 18 metabolic markers were compared in our analysis, it is likely that the difference found in thyroxin levels between groups is spurious.

Regarding self-body image, it is reasonable to state that the BMI is the best determinant of the current self-body image perception (*Kaufer-Horwitz et al., 2006*). In our sample, BMI was a significantly weaker predictor of current self-body image assessment in SS women as compared to NS women, and this relationship remained significant even after correction for possible confounding social variables. In addition, obese SS women were less likely to perceive themselves as obese as compared to NS women.

*Holdsworth et al. (2004)* found that Senegalese women believed that being overweight was socially preferred, but being obese was associated with a greater risk of disease. In a longitudinal investigation, *Lynch et al. (2009)* found that obese women who did not

perceive themselves as obese gained more weight over a course of 13 years. *Powell et al. (2010)* showed that misperception about body size is prevalent among obese adults, particularly among ethnic minorities. They emphasized that this phenomenon may be linked with an overestimation of health and underestimation of risk, which may lead to a lower utilization of the health care system and inadequate physician counseling. Taken together, all these issues could enhance the risk of mortality among those with impaired self-body image in the long-term. It is possible that SS women perceive themselves as small as compared to NS women. Since body weight is possibly their parameter of choice to assess their body size, owing to its simplicity, they may not perceive themselves as have "excess weight" when they weigh approximately 56 kg, which is the average weight for a 149-cm tall woman to be considered overweight.

This study has some limitations. First, we used a non-probabilistic convenience sample, which decreases our ability to extrapolate our findings. Second, we used bioimpedance analysis to assess body composition, which is not a gold standard method, such as deuterium. Third, we used a single scale to assess body image, which may weaken our findings regarding this characteristic.

## CONCLUSIONS

In conclusion, the present study showed that overweight/obese, shantytown-dwelling SS women present with more central adiposity and impaired self-body image perception than NS women, and BMI is a weaker predictor of current self-body-image perception. These findings could partially explain the higher mortality rates found in SS adults. Further longitudinal investigations, assessing these characteristics in mortality or cardiovascular events are needed.

### Funding
The authors received no funding for this work.

### Competing Interests
The authors declare that they have no competing interests.

### Author Contributions
- Nassib Bezerra Bueno conceived and designed the experiments, performed the experiments, analyzed the data, wrote the paper, prepared figures and/or tables.
- Telma Toledo Florêncio conceived and designed the experiments, performed the experiments, contributed reagents/materials/analysis tools, reviewed drafts of the paper.
- Fabiana Albuquerque Cavalcante performed the experiments, wrote the paper.
- Isabela Lopes Lins performed the experiments, wrote the paper.
- Ana Grotti Clemente performed the experiments, analyzed the data, wrote the paper.
- Ana Lydia Sawaya conceived and designed the experiments, contributed reagents/materials/analysis tools, reviewed drafts of the paper.

## Human Ethics

The following information was supplied relating to ethical approvals (i.e., approving body and any reference numbers):

Data were collected from all participants after they provided written informed consent. The study was approved by the Ethical Research Committee of the Federal University of São Paulo, number 275184.

## Data Deposition

The raw data has been supplied as Supplemental Dataset Files.

## Supplemental Information

Supplemental information for this article can be found online at http://dx.doi.org/10.7717/peerj.2547#supplemental-information.

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
