# Peer review of "Higher central fat and poor self-body image in short-stature overweight/obese women living in Brazilian shantytowns"

_PeerJ, doi:10.7717/peerj.2547_

## Round 0.1 · original submission · Major Revisions

Please carefully consider all the comments raised up by the reviewers. Specially, the Results and Discussion headings should be clearly improved, following the modifications they suggested.

Reviewer 1 ·

Basic reporting

Discussion of abstract is long, and introduces specific new information outside the scope of the data collected; this extra information should be removed.

Paper should include information on the recruitment process: how many screened, how many non-eligible and for what reasons, how many eligible but declined to participate, etc.

Writing in the results section (lines 164-182) is often unclear. For example, for some sentences it is difficult to keep track of which model is being discussed.

Table 2. Not clear if these are adjusted or unadjusted P values. Need a table footnote.

Lines 47-49: reference for this?

Lines 54-61 in Introduction: Introduction would be strengthened by addressing the differences between linear growth among children and stature as an adult in terms of determinants and also possible effects.

Table 2. Seems odd that a difference in body fat % of 1% would be statistically significant (p=0.06). Is this adjusted or unadjusted? Was this t-test or nonparametric? Maybe useful to show the full distribution of values.

Table 3. Why is % body fat not presented in the body weight adjusted analysis?
Not clear why the results of table 3 are described as “opposite” in the results section. In both Tables 2 (unadjusted?) and 3 (adjusted for body weight?), fat mass and lean mass are greater in NS women compared to SS.

While it is understood that “the mechanisms that lead to increased cardiovascular risks in SS 189 adults are not fully understood”, the paper would benefit from a stronger explanation of the possible mechanisms. In particular, is there some sort of ‘biological programming’ that happens following early life exposures (authors seem to suggest this), or could the association be completely explained by confounding?

Experimental design

The convenience sample is a limitation, as noted by the authors. The design would be strengthened by collecting a representative sample, or at least by including SS and NS groups that are not overweight.

Line 116: Note that ubilical point is not always the same location as the midpoint between iliac crest and lower rib, which is recommended for waist circumference. This difference may overestimate waist circumference. This should be mentioned in the discussion.

Validity of the findings

Regarding the difference in total T-4: Because there are 18 comparisons in this table, with alpha=0.05, it would not be surprising if 1 is a false positive. Probably worthwhile to mention this in discussion.

Discussion: “In our cross-sectional study, we found that overweight/obese SS
190 women had relatively more fat mass and less lean mass than their non-short counterparts.” This does not seem consistent with the data presented in Table 2 or Table 3. Is there a typo in the table?

Discussion: “In our sample, we did not find any differences in the WHR between the
219 NS and SS groups, as was also reported by Florêncio et al. (2007). This was probably because 220 our NS group had greater WC than the SS group.”
In theory the NS group would also have greater hip circumference, so it is not clear why greater WC would explain this.

Additional comments

Another reference which may be of use:

A century of trends in adult human height.
NCD Risk Factor Collaboration (NCD-RisC).
Elife. 2016 Jul 26;5. pii: e13410. doi: 10.7554/eLife.13410.

Reviewer 2 ·

Basic reporting

This is study supports that short-stature women have high central fat and distortion on self-body-image. This finding become relevant for understanding the self-perception and increase cardiovascular risk in short stature women.
The paper has clear statements, with correct language use. The topic is socially relevant, original and can be the source for future community interventions; however, the authors should consider important changes to improve the methodology and interpretation of the results.

Experimental design

Well designed with defined contrast groups.

Validity of the findings

The interpretation of results should be analyzed in deep (check the following box).

Additional comments

The abstract should be less extend, and consider a better description of results, for instance include the effect sizes and not only p-values. The abstract discussion should be focused on the current research, there are a lot of irrelevant information. Meanwhile, the discussion on the body has better structure and discusses social risks during early life that can have effect during adulthood.

The concept that low T4 with normal TSH levels could be related with SS findings is just merely speculation with no bases. The TSH values shows three high outliers in the normal stature women, and they did not correlate with the T4 levels.
It is common the T4 can show wide serum level fluctuations, therefore, the TSH is the most reliable study to analyze the thyroid function. You should discuss the T4 difference upon physiological bases. The TSH did not show difference among the two groups, hence the T4 contrast is spurious.

Table 1. I suggest to use the term “frequency” instead of “n”.
Table 3. Please describe at bottom which variables were used for adjustment? This table does not make sense. How possible the lean mass be more than 140 kg and fat mass more than 70 kg in women who weight a mean of 80 kg?
Table 4. The U Mann-Whitney test does not have a place here. You have continuous variables and you can deal with log transformed variables, or in any case, the Student t test can correct the p values by variance differences.
Figure 1. The authors should ask their self: What do I intend to show with this figure? Perhaps changing the axes and using non-linear trends or box-plots can give a better idea of their message.
Figure 2 is irrelevant, can be described in text.

---

## Round 0.2 · accepted · Accept

I think the authors have properly answered the questiosn raised by the reviewers. The manuscript has clearly improved with the new modifications.